# Parameterization of 3D cloud geometry and a neural-network-based fast forward operator for polarized radiative transfer

Anna Weber<sup>1</sup>, Gregor Köcher<sup>1</sup>, and Bernhard Mayer<sup>1</sup>

<sup>1</sup>Meteorologisches Institut, Ludwig-Maximilians-Universität München, Munich, Germany

**Correspondence:** Anna Weber (Weber.Ann@physik.uni-muenchen.de)

**Abstract.** Clouds generally have a complex three-dimensional geometry. However, realistic three-dimensional radiative transfer simulations of clouds are computationally expensive, so most retrievals of cloud properties assume one-dimensional clouds, which introduces retrieval biases. In this work, a fast forward operator for polarized 3D radiative transfer in the visible wavelength range is presented. To this end, a new approximation for 3D radiative transfer, the InDEpendent column local half-sphere ApproXimation (IDEFAX), is introduced. The basic idea behind this approximation is similar to the independent column approximation assuming plane-parallel clouds. However, every column is approximated by an independent field of 3D halfspherical clouds instead of a plane-parallel homogeneous cloud. This field of half-spherical clouds is defined by the local cloud surface orientation angles and the cloud fraction. Thus, the IDEFAX has only three more parameters compared to the planeparallel approximation. To obtain a fast forward operator, artificial neural networks were trained for both the plane-parallel and the half-spherical cloud assumptions. The IDEFAX as well as the neural network forward operators were validated against polarized 3D radiative transfer simulations with MYSTIC for low-level Arctic mixed-phase clouds using a realistic cloud field simulated with the WRF model. The use of the IDEFAX significantly improves the representation of 3D radiative effects in the simulated radiance fields compared to the plane-parallel independent column approximation. Due to the implementation of the forward operator with neural networks, the computation time for both approximations is comparable and about five orders of magnitude faster than full 3D radiative transfer simulations for the shown example. The introduced neural network forward operators are constructed to be used in retrievals of cloud properties with the specMACS instrument. However, the methods are also applicable to other measurements in the visible wavelength range as well as to model data.

## 1 Introduction

In general, clouds have a complex three-dimensional geometry and cloud properties vary vertically and horizontally. This complex 3D structure leads to different 3D radiative effects due to horizontal transfer of radiation. The finite extent of clouds allows for example for the escape of radiation through the cloud sides on the one hand and side illumination effects on the cloud side oriented towards the incoming solar radiation on the other hand. In addition, variations of cloud top inclination relative to the sun lead to different projected areas and thus radiation enhancements or reductions including shadowing effects. Three-dimensional radiative effects can influence radiation on a local level as well as affect larger areas. They can lead to both roughening and smoothing of the brightness field (e.g. Marshak et al., 1995; Várnai and Marshak, 2003). The causes

and impacts of those different 3D effects were for example studied quantitatively by Várnai and Marshak (2003). 3D radiative effects may also include effects due to spatial variations of microphysical properties of clouds like the cloud droplet or ice crystal size. This work, however, focuses on 3D radiative effects due to cloud geometry.

Realistic three-dimensional radiative transfer simulations are very expensive, so 1D plane-parallel clouds in the independent column approximation (ICA) are assumed for most operational retrievals of cloud properties and also with model data. In the ICA, every pixel is treated independently and represented by a plane-parallel cloud. However, Di Girolamo et al. (2010) found that the plane-parallel assumption was consistent with only 24% of the observed reflectances from satellite instruments for solar zenith angles smaller than 60° and with even less for larger solar zenith angles. Due to this assumption and the neglection of 3D radiative effects, there are significant biases in typical bispectral retrievals (Nakajima and King, 1990) of optical thickness (e.g. Várnai and Marshak, 2002) and effective radius (Marshak et al., 2006; Vant-Hull et al., 2007).

Polarization is in general less influenced by three-dimensional radiative effects since it is dominated by single scattering. However, 3D radiative effects can still have a non-negligible effect on polarization measurements. Cornet et al. (2010), for example, studied the influence of 3D cloud geometry on polarized reflectances for cirrus clouds and found non-negligible biases also for the polarization signal for high spatial resolutions of 80 m as well as for low resolutions of 10 km.

40

50

There are different approaches to correct retrieval results or simulated radiances for three-dimensional radiative transfer effects. Alexandrov et al. (2024) developed a correction based on the aspect ratio of the observed cloud to correct for retrieval errors due to 3D effects in the cloud optical thickness values retrieved from MODIS measurements. Scheck et al. (2018) introduced a cloud top inclination correction for simulated radiances of optically thick clouds. Meyer et al. (2022) used a machine learning approach to correct 1D radiative transfer simulation results for numerical weather models for three-dimensional radiative effects. They achieved to correct 70% to 80% of the three-dimensional radiative effects missed by one-dimensional radiative transfer simulations. For the thermal wavelength range, Schäfer et al. (2016) and Fielding et al. (2020) showed that the sub-grid scale three-dimensional cloud geometry and the resulting three-dimensional radiative effects in numerical weather prediction and climate models can be parameterized through the cloud edge length. However, we are not aware of any similar approximation of 3D cloud geometry for the computation of radiances in the visible wavelength range.

In this work, a new parameterization of 3D cloud geometry for polarized 3D radiative transfer in the visible wavelength range is presented and applied to obtain a fast forward operator. The new InDEpendent column local half-sphere ApproXimation (IDEFAX) is based on the ICA. However, it uses a new simplified cloud model instead of the plane-parallel approximation. The 3D cloud geometry of an individual column is approximated by a field of half-spherical clouds that are defined by two local surface orientation angles and the cloud fraction as visualized in Figure 3. A spherical cloud shape as approximation to real clouds was already proposed by Davis (2002). The addition of only three parameters in the IDEFAX compared to the plane-parallel ICA makes it simple to implement and simple enough to tabulate simulated radiances. The forward operator for polarized 3D radiative transfer provides the Stokes vector components I, Q, and U and accounts for 3D cloud geometry using the new IDEFAX. Instead of a look-up-table, artitifical neural networks were trained as forward operators for both, the IDEFAX as well as the plane-parallel ICA. They provide fast radiative transfer simulation results for the Stokes vector components I, Q, and U with small memory requirements in comparison to a look-up-table. To validate the IDEFAX and

the neural network forward operators, a realistic cloud field consisting of low-level Arctic mixed-phase clouds was simulated using the Weather Research and Forecasting (WRF) model (Skamarock et al., 2019). Real three-dimensional polarized radiative transfer simulations of the cloud field performed with the Monte Carlo 3D radiative transfer solver MYSTIC (Mayer, 2009) were then compared to simulated radiances from the IDEFAX and the plane-parallel ICA, and the respective neural network forward operators.

The forward operator was constructed for retrievals using measurements of the polarization-resolving cameras of the spec-MACS instrument. However, it is in principle also applicable to other measurements or to model data. specMACS (Ewald et al., 2016; Weber et al., 2024) is a hyperspectral and polarized imaging system that is operated in a downward-looking perspective on board of the German research aircraft HALO. Its 2D RGB polarization-resolving cameras measure the first three components of the Stokes vector (I, Q, and U) in red, green, and blue color channels with a high spatial resolution of about 10m at a typical flight altitude of 10km. The center wavelengths of the color channels are about 620 nm, 550 nm, and 470 nm. The cameras have a large field of view and allow for the application of multi-angle polarimetric retrievals. Further information about the polarization-resolving cameras can be found in Weber et al. (2024). Cloud 3D geometry can be derived from the measurements using the stereographic reconstruction method by Kölling (2020) which uses contrasts in total intensity measurements. Especially for measurements with large solar zenith angles as during the HALO- $(\mathcal{AC})^3$  measurement campaign (Wendisch et al., 2024) in the Arctic, three-dimensional radiative effects become very important and retrievals need to consider the effects of 3D cloud geometry.

The paper is organized as follows. The cloud simulations conducted with the WRF model are explained in Section 2 followed by an overview of the radiative transfer model MYSTIC in Section 3. Next, the developed simplified cloud model for the IDEFAX is described in Section 4 and the neural network forward operator setup, training, and results are given in Section 5. The IDEFAX and the neural network forward operator are validated using 3D radiative transfer simulations for the cloud field simlated with the WRF model in Section 6. Finally, the findings of this work are discussed and summarized in Section 7.

### 2 WRF simulations

A realistic cloud field was simulated to test different approximations of 3D cloud geometry for polarized radiative transfer and to validate them against full 3D radiative transfer simulations. The simulations were performed with the Weather Research and Forecasting (WRF) model version 4.6 (Skamarock et al., 2019) using four two-way nested domains with a nesting ratio of 5:1. For this study only data from the innermost domain are used. The inner domain extends from 4.74°E to 7.24°E and from 79.22°N to 79.67°N (25 km x 25 km) with a horizontal grid spacing of 100 m and 200 vertical levels. The simulation covers the time period from 0 UTC to 15 UTC on 2022-04-01. Initial and boundary conditions were provided by ERA5 reanalysis data (European Centre for Medium-Range Weather Forecasts, 2019) at 0.25° grid spacing. Important simulation physics options include the 2-moment bulk microphysics scheme by Morrison et al. (2009), the Unified Noah Land Surface Model (LSM; Tewari et al., 2004), the Mellor-Yamada Nakanishi Level 2.5 planetary boundary layer (PBL) scheme (MYNN2; Nakanishi and Niino, 2006, 2009; Olson et al., 2019), and the rapid radiative transfer model for general circulation models as the radiation

scheme (RRTMG; Iacono et al., 2008). For a complete list of the simulation settings, please refer to the WRF namelist provided in the supplement.

Next, the output of the WRF model was converted to input for the radiative transfer model where clouds are defined by their liquid or ice water content and respective effective radius. Liquid water content (LWC) and ice water content (IWC) are directly included in the WRF model output. All liquid hydrometeors were interpreted as liquid clouds and all frozen hydrometers as ice clouds for the radiative transfer simulations. The liquid cloud effective radii  $r_{\rm eff,wc}$  and ice cloud effective radii  $r_{\rm eff,ic}$  needed for radiative transfer were computed from the liquid water content and ice water content and the respective number concentrations from the model output following the equations by Martin et al. (1994). Different methods for computing ice effective radii from the model output were tested and showed similar results. From the IWC, LWC, and liquid and ice effective radii fields, liquid and ice optical thickness fields were computed. In the end, the effective radii were set to typical values of  $10 \mu m$  for liquid and  $50 \mu m$  for ice clouds throughout the cloud field, and LWC and IWC scaled accordingly keeping the optical thickness conserved. By assuming constant effective radii, additional effects due to variations of the effective radius were excluded. Generally, the variation of radiances due to varying effective radii is however small in the visible wavelength range.

Fig. 1 displays the part of the obtained cloud field on 2022-04-01 at 3 UTC that is visible in the radiative transfer simulations in Sections 3 and 6. The upper row shows horizontal fields of liquid (a) and ice optical thickness (b), the bottom row vertical cross sections of the cloud field along the southern most edge at a latitude of 79.22° north for the LWC (c) and IWC (d). On the simulated day on 2022-04-01, there was a marine cold air outbreak in the Fram strait. During a cold air outbreak, cold and dry polar air masses are transported southwards. When they pass the marginal sea ice zone and reach open ocean, convection sets in and clouds are formed. They usually first organize into cloud streets and later on develop more cellular structures (Fletcher et al., 2016; Papritz and Spengler, 2017). This is also visible in the simulated cloud field in panels (a) and (b) where the optical thickness shows cloud streets oriented roughly in north-south direction. The clouds have cloud top heights smaller than 1000 m, and liquid water is located more towards the top of the clouds while the ice is concentrated in the lower part as can be seen in panels (c) and (d).

### 3 Radiative transfer model






The radiative transfer simulations were performed with the radiative transfer model libRadtran (Mayer and Kylling, 2005; Emde et al., 2016) using the Monte Carlo solver MYSTIC (Mayer, 2009) which allows for full 3D radiative transfer simulations including polarization (Emde et al., 2010). In Monte Carlo radiative transfer simulations, photons are traced through the atmosphere, where they can be scattered by clouds, aerosols, and molecules, reflected by the Earth's surface, or absorbed, until they reach the detector. The chosen viewing geometry is a typical viewing geometry of the polarization-resolving cameras of the specMACS instrument during the HALO-(AC)<sup>3</sup> campaign and corresponds to the measurements on 2022-04-01 at 10:18 UTC. The part of the cloud field that is covered by the instrument field of view for the chosen viewing geometry is indicated by the black dashed lines in Figure 1. Radiative transfer simulations for the specMACS instrument were already used by Volkmer et al. (2024) but their work only included liquid water clouds. The solar zenith angle for the radiative transfer simulations was

**Figure 1.** WRF cloud field. (a,b) Optical thickness for liquid and ice clouds, respectively. (c,d) Vertical cross sections of LWC and IWC along latitude 79.22° north. The dashed lines indicates the part of the cloud field that is covered by the field of view of the instrument and visible in the radiative transfer simulations in Sections 3 and 6.

75.6°. Concerning cloud optical properties, phase functions according to Mie theory (Mie, 1908) were used for liquid clouds and the ice optical properties by Yang et al. (2013) for the aggregate consisting of eight columns with moderate crystal roughness for ice clouds. This habit was chosen as compromise between pristine plates and columns and more complex aggregates. The simulations were done for a US standard atmosphere and using the ocean BRDF by Cox and Munk (1954a, b) with wind speed 10 m/s and 0° wind direction which means wind coming from the north. The obtained Stokes vectors for all simulations were rotated into the scattering plane and converted to reflectivity via:



$$R_I = \frac{\pi I}{E_0 \cos(\text{sza})} \qquad \text{and} \qquad R_Q = \frac{\pi Q}{E_0 \cos(\text{sza})} \tag{1}$$

where I and Q are the first and second Stokes vector components describing total intensity and linear polarization, sza is the solar zenith angle, and  $E_0$  the extraterrestrial solar irradiance.

Fig. 2 shows 3D radiative transfer simulations for the green color channel of specMACS and the WRF cloud field as described above in panels (a) and (c) and the corresponding real measurement in panels (b) and (d). The radiative transfer simulations have the same spatial resolution of about 10 m as the measurements. The upper row displays total reflectance  $R_I$  and the bottom row the polarized reflectance  $R_Q$ . In the polarization signal, the cloudbow is visible as a minimum in  $R_Q$  in the lower right corner. It is formed by single scattering by liquid droplets. The simulation in general looks smoother than the mea-

Figure 2. Full 3D radiative transfer simulation using the WRF cloud field (a, c) and specMACS measurement on 2022-04-01 10:18 UTC (b, d) for the green color channel. (a, b) show reflectivity  $R_I$ , (c, d) the reflectivity for the Q-component of the Stokes vector  $R_Q$ . The dashed lines indicate the scattering angles.

surement which is due to the lower spatial resolution of the WRF cloud simulations (100 m) compared to the high-resolution measurements (10 m). The overall structure of the simulated clouds and also the absolute values of the simulated  $R_I$  and  $R_Q$  are realistic compared to the measurements. The bipolar structures in the polarization signal of the measurements in the upper left and lower right corners in panel (d) are calibration artifacts due to the more uncertain polarization calibration in the corners of the sensor.

#### 4 Simplified cloud model


With the WRF simulations and the radiative transfer model, it was possible to test different simplified cloud models to approximate 3D cloud geometry for polarized radiative transfer (see Figure 3). Realistic three-dimensional clouds have a complex cloud surface geometry as visualized in Fig 3(a). The basic idea was to find a simplified description of the 3D clouds that covers most of their 3D characteristics but is simple enough to be defined by a small number of parameters such that radiances, respectively Stokes vectors, for the approximation could be tabulated.

In the case of the plane-parallel approximation (Fig. 3(c)), a cloud is completely defined by its cloud top height (cth), cloud base height, optical thickness  $\tau$ , and effective radius ( $r_{\rm eff}$ ). To further reduce the number of parameters, the geometrical thickness d of the cloud, and thus the cloud base height, was parameterized through the optical thickness of the cloud using the

**Figure 3.** Definition of different cloud geometries and parameterizations. (a) Full three-dimensional clouds. (b) Half-spherical clouds. (c) One-dimensional clouds.

adiabatic cloud model (e.g. Grosvenor et al., 2018):





$$d = \sqrt{\frac{4}{3}} \frac{\rho_{\rm w} r_{\rm eff}}{f_{\rm ed} C_{\rm w}} \tau \tag{2}$$

Here,  $\rho_{\rm w}$  is the density of liquid water. The adiabaticity is set to a typical value of  $f_{\rm ad}=0.3$  (Ishizaka et al., 1995; Merk et al., 2016) and the condensation rate to  $C_{\rm w}=2.5\times 10^{-6}~{\rm kg/(m^3m)}$  (Min et al., 2012). A derivation of the formula based on the adiabatic cloud model can be found in Appendix A. For the computation of the geometrical thickness from the optical thickness, the constant effective radius of the simulated cloud field (see Section 2) is applied. Hence, optically thick clouds are also geometrically thicker and a one dimensional cloud can be described only by its cloud top height, optical thickness, and effective radius. Although the geometrical thickness of a cloud affects the Rayleigh scattering within a cloud, the sensitivity of simulated Stokes vectors to the cloud geometrical thickness is comparably small which allows for the described approximation of the cloud geometrical thickness through the cloud optical thickness.

In a next step, different cloud geometries were tested as approximations to the real three-dimensional clouds. This included box clouds to account for the finite cloud size, tilted clouds to approximate the effect of cloud top inclination, and half-spherical clouds that include finite cloud size as well as cloud top inclination. In all cases, the tested approximations of 3D cloud geometry are combined with the ICA and thus applied independently to every pixel. This means that, for example in the case of one-dimensional clouds, for every pixel an independent simulation is performed with a plane-parallel cloud located at the cloud top height and with the optical thickness and effective radius of that pixel. An explanation how these parameters can be obtained for every pixel from measurement or model data is described in detail in Section 6. In this work, the simulated pixel size was about 10 m since this is the spatial resolution of the measurements of specMACS. For other applications, the pixel size could be chosen differently. A comparison of simulated Stokes vectors obtained with the different approximations against full 3D radiative transfer simulations of the realistic cloud field from the WRF model showed, that neither the finite cloud size (represented by the box clouds) nor the surface orientation (represented by the tilted clouds) alone are sufficient to capture the basic 3D radiative effects.

The cloud model that showed the best agreement with the full 3D cloud field was a field of half-spherical clouds as shown in Fig. 3(b). We call this new approximation InDEpendent column local half-sphere ApproXimation (IDEFAX). The half-spherical clouds are, in addition to cloud top height, optical thickness, and effective radius, described by a surface orientation

**Figure 4.** Basic geometry of the IDEFAX. The grey lines indicate a single column of a complex cloud. This column is approximated by the dashed half-spherical cloud which has the same surface orientation and cloud top height as the real cloud in the considered column.

zenith (oza) and azimuth angle (oaz) and the cloud fraction ( $f_{cloud}$ ). Their geometrical size is defined through the optical thickness as above. Here, the surface orientation zenith angle is the angle between the local surface normal of the cloud surface at a given point on the cloud surface and the vertical. The surface orientation azimuth angle is the angle between the principal plane, which contains the sun vector and the vertical, and the plane formed by the surface normal and the vertical (see also Figure 3). Surface orientation angles can directly be derived from the cloud geometry of model or measurement data, e.g. by 185 fitting a plane through the 3D points describing the cloud surface in neighboring pixels as in Scheck et al. (2018) or, in case for application to specMACS data, from the stereographic retrieval of cloud 3D geometry by Kölling (2020). In the ICA, individual pixels are treated independently and thus the real 3D cloud for every pixel is approximated by an independent field of half-spherical clouds. For that, a certain target point on a half-sphere is defined by the surface orientation azimuth and zenith angles of the considered pixel and the half-sphere is placed such that the target point is located at the given cloud top height, 190 see Fig 4. This half-sphere is part of a field of half-spheres with a given cloud fraction. The cloud fraction is the ratio of the number of cloudy pixels to the total number of pixels and thus contains non-local information. It can be computed from measurement or model data by applying a cloud mask. Hence, a cloud field consisting of half-spherical clouds as described above is constructed such that the number of cloudy pixels within the cloud field corresponded to the given cloud fraction. The cloud fraction has an upper limit that is less than unity since half-spheres cannot completely fill the model domain. An isolated half-spherical cloud not embedded into a cloud field showed less agreement with full 3D radiative transfer simulations than 195 the field of half-spherical clouds defined through the cloud fraction. Clouds in MYSTIC are defined on a regular grid. For the half-spherical clouds, the stepsize of the grid boxes building up a cloud was chosen such that the optical thickness per step was less than 0.5 to avoid artifacts from the steps but also minimize the required amount of memory. This internal stepsize is independent of the simulated pixel size.

In the case of mixed-phase clouds, the clouds are defined by their total optical thickness and the ice fraction which is here defined as the ratio of the ice optical thickness to the total optical thickness. They are simulated as clouds with homogeneously mixed liquid and ice in the parameterization.

# 5 Neural network forward operator








Although only three extra parameters are required for the approximation of 3D cloud geometry through half-spherical clouds with the IDEFAX, a look-up table that includes radiative transfer simulation results for all necessary parameters would be prohibitively large because 11 parameters (see Table 1) are already needed for simple one-dimensional clouds. The computation as well as interpolation within the look-up table would be expensive and the memory requirements would be high. Therefore, we decided to replace the look-up table by a simple feed forward neural network. Artificial neural networks allow for very fast inference with little storage space needed. In addition, neural networks are by definition differentiable (if differentiable activation functions are used) which allows for the direct computation of gradients. This can be advantageous if the neural network is applied in retrievals that use optimization. Separate neural networks were trained for one-dimensional clouds in the classical ICA and the half-spherical clouds of the IDEFAX.

Training data was computed using MYSTIC by randomly sampling all input parameters within their boundaries. The WRF simulations are independent of the generation of training data and the neural network training and were only used to validate the IDEFAX and the neural network forward operators in Section 6. Table 1 summarizes all input parameters that were used for the training of the neural networks. They include geometrical information like the solar zenith angle, viewing zenith and azimuth angles, sensor height and cloud top height as well as information about cloud microphysical properties such as the total optical thickness, ice fraction, and effective radius and variance of a cloud. The cloud microphysical properties are for measurement data in most cases unknown and derived by retrievals. Finally, it includes the ocean surface with the ocean BRDF by Cox and Munk (1954a, b) as lower boundary defined by wind speed and wind direction. Compared to the other parameters, the exact values of wind speed and wind direction describing the ocean surface are less relevant for radiative transfer simulations of clouds. For the IDEFAX, cloud surface orientation zenith and azimuth angles as well as the cloud fraction are additional parameters. The azimuthal angles are defined relative to the principal plane. All simulations were performed for a US standard atmosphere and cloud base height was parameterized through the optical thickness as discussed in Section 4. Moreover, the ice cloud effective radius was assumed to be 50 µm and the aggregate of eight columns with moderate surface roughness by Yang et al. (2013) used as ice crystal habit. Depending on the application, ice crystal effective radius and habit could be added as additional parameters in the future. The ranges for all input parameters were chosen such that they cover the typical ranges measured by an aircraft instrument. Due to the focus on polarization, optical thickness values were only sampled up to an optical thickness of 8. The polarization signal saturates latest at an optical thickness around 5 such that the full range is already covered for the chosen optical thickness boundaries. For other applications, the boundaries of the input parameters could be adjusted. All input parameters were normalized within their boundaries for the neural network training.

Output quantities of the neural networks are the Stokes vector components I, Q, and U converted to reflectivity. In the following, the results of the neural network setup and training for the green color channel (which has a center wavelength close to 550nm) of one polarization-resolving camera of the specMACS instrument are shown. The results for the other visible color channels of the specMACS instrument are similar and thus potentially also the results for visible wavelength channels of other instruments. The radiative transfer simulations for the training data were done for a spectrum covering the wavelength

**Table 1.** Input parameters, their abbreviations, and ranges. The last three parameters are for half-spherical clouds only.

| Parameter                               | Abbreviation    | Minimum           | Maximum             |
|-----------------------------------------|-----------------|-------------------|---------------------|
| Solar zenith angle                      | sza             | 0°                | 80°                 |
| Viewing zenith angle                    | vza             | $0^{\circ}$       | 70°                 |
| Viewing azimuth angle                   | vaz             | $0^{\circ}$       | 180°                |
| Sensor height                           | zout            | $8000 \mathrm{m}$ | $15000 \mathrm{m}$  |
| Cloud top height                        | cth             | 500m              | $13000 \mathrm{m}$  |
| Total optical thickness                 | au              | 0                 | 8                   |
| Ice fraction                            | $f_{ m ice}$    | 0                 | 1                   |
| Liquid cloud effective radius           | $r_{ m eff,wc}$ | 1µm               | $40 \mu \mathrm{m}$ |
| Liquid cloud effective variance         | $v_{ m eff,wc}$ | 0.01              | 0.32                |
| Surface wind speed                      | u10             | $1 \mathrm{m/s}$  | $15 \mathrm{m/s}$   |
| Surface wind direction                  | uphi            | $0^{\circ}$       | 360°                |
| Cloud surface orientation zenith angle  | oza             | 0°                | 60°                 |
| Cloud surface orientation azimuth angle | oaz             | -180°             | 180°                |
| Cloud fraction                          | $f_{ m cloud}$  | 0                 | 1                   |

range from 380nm to 690nm in 10nm steps with a standard deviation of the simulation results of 4% per wavelength and then integrated to the green wavelength channel by applying the spectral response function of the polarization-resolving cameras. In total, 20 million random samples were computed for the one-dimensional clouds with 11 input parameters and 40 million random samples for the more complex IDEFAX with 14 input parameters. These were obtained by performing individual radiative transfer simulations for randomly chosen sets of values for the input parameters listed in Table 1. Two million samples of the simulated training data were used for testing.




Next, both neural networks were trained and hyper-parameters tuned until a good setup was found. Training and tuning included testing different network sizes with different numbers of hidden layers and numbers of total parameters, batch sizes, learning rates, activation functions as well as parameter transformations (similar to Scheck (2021) and Baur et al. (2023)). In the following, the results for the best models will be shown. For both, the one-dimensional and the IDEFAX models, a weighted mean squared error was used as loss. Q and U were given ten times the weight of I to account for the different orders of magnitudes of I, Q, and U. In addition, the loss was multiplied by a factor of 1000 to avoid vanishing gradient problems. In general, the dependence of the cloudbow on effective radius and variance was difficult for the neural networks to learn since the cloudbow is a strongly non-linear feature. The addition of the scattering angle as an input parameter improved the performance and smaller loss values were obtained. The scattering angle was computed from the solar and viewing geometry. In addition, the optical thickness was transformed logarithmically and a square root transformation was applied to the output Stokes vectors

**Table 2.** Error statistics for the best neural networks for one-dimensional clouds and half-spherical clouds for I, Q, and U computed using the test data.

| Neural network        | Parameter | Bias                  | RMSE                 | MAE                  |
|-----------------------|-----------|-----------------------|----------------------|----------------------|
| 1D clouds             | I         | $2.7\times10^{-4}$    | $5.0\times10^{-3}$   | $3.5 \times 10^{-3}$ |
|                       | Q         | $-7.9\times10^{-6}$   | $1.4\times10^{-3}$   | $7.6\times10^{-4}$   |
|                       | U         | $1.1\times10^{-4}$    | $9.7\times10^{-4}$   | $6.3\times10^{-4}$   |
| Half-spherical clouds | I         | $-4.8 \times 10^{-4}$ | $2.8 \times 10^{-2}$ | $1.2 \times 10^{-2}$ |
|                       | Q         | $8.7\times10^{-5}$    | $9.2\times10^{-3}$   | $1.7\times10^{-3}$   |
|                       | U         | $8.1\times10^{-5}$    | $1.3\times10^{-3}$   | $6.9\times10^{-4}$   |

as in Scheck (2021). Moreover, a sine-cosine transformation was applied to the wind direction and surface orientation azimuth angles to avoid the jump from  $360^{\circ}$  to  $0^{\circ}$ . This means that the viewing azimuth angle was described by  $\cos(vaz)$  and  $\sin(vaz)$ .





After hyper-parameter tuning for the one-dimensional clouds, a neural network with 8 hidden layers with 64 nodes per layer, corresponding to 30000 parameters in total, gave the best results. More parameters resulted in overfitting while less parameters were not able to capture all the details of the Stokes vector fields. Neural networks with less or more hidden layers showed worse results compared to the neural network with 8 hidden layers. Moreover, the exponential linear unit (elu) activation function and a linear output activation function were used in the best model setup. Other activation functions like the cheap soft unit (csu) or the hyperbolic tangent (tanh) gave worse results. Finally, a batch size of 1024 and the Adam optimizer with a learning rate of  $1.0 \times 10^{-4}$  was applied in the network training of the best model. It was trained for 2000 epochs using an early stopping routine to get the most accurate results. Error statistics for the best model for one-dimensional clouds including bias, root mean squared error (RMSE), and mean absolute error (MAE) for I, Q, and U are summarized in Table 2. The statistics were computed from the test data. The obtained biases are two to three orders of magnitude smaller than typical signal levels. In addition, the errors are significantly lower than the measurement uncertainties of typical polarization resolving instruments which are on the order of a few percent (e.g. Weber et al., 2024).

The neural network using half-spherical clouds with the IDEFAX was trained similarly to the network for plane-parallel clouds. Best results were found for a neural network with 6 hidden layers and 97 nodes per layer resulting in a total number of parameters of about 50000. As for the 1D clouds, the elu activation function and a linear output activation were used and the model trained for 2000 epochs using early stopping. In addition, a batch size of 1024 and learning rate of  $1.0 \times 10^{-4}$  for the Adam optimizer showed most accurate results. The obtained prediction errors of the neural network for the half-spherical clouds in the IDEFAX computed from the test data are also summarized in Table 2. The errors, especially for I, are larger compared to the neural network for 1D clouds. However, they are still smaller than typical measurement uncertainties. The reason for this is that the half-spherical cloud geometry introduces more non-linearities, in particular for the intensity, and is thus much more difficult for a neural network to learn.

For fast emulation, the fornado module by Scheck (2021) was used which is optimized for predictions of small neural networks. In addition, it allows the computation of adjoints based on the Tapenade tool for automatic differentiation (Hascoet and Pascual, 2013) which can be used for optimization routines in retrievals. The full 3D simulations of the images displayed in Figure 6 took on the order of several days on an Intel® Xeon(R) W-2245 CPU @ 3.90GHz processor. The plane-parallel ICA simulations with MYSTIC needed about a day and the IDEFAX simulations with MYSTIC on the order of a few days. In contrast, the predictions with the neural networks in Figure 8 took both only on the order of seconds with slightly larger computation times for the IDEFAX than for the plane-parallel ICA. Thus, a total speed up of the computation time by a factor of about 10<sup>5</sup> can be achieved by using the neural network forward operators instead of performing polarized Monte Carlo radiative transfer simulations.

# 6 Validation of the IDEFAX and the neural network forward operator






Both, IDEFAX and the neural network forward operators were validated against full 3D radiative transfer simulations with MYSTIC for the realistic cloud field obtained from the WRF simulations. The radiative transfer simulations were performed for a wavelength of 550nm which is close to the center wavelength of the green color channel of the specMACS polarization-resolving cameras the neural networks were trained for and used the settings described in Section 3. The radiances obtained from the 3D radiative transfer model were then compared to radiances for the same cloud field computed with the IDEFAX and the plane-parallel ICA. In addition, the comparison was repeated with the neural network forward operators for the IDEFAX and plane-parallel clouds for the green color channel of specMACS to also validate the forward operators.

The input parameters defining the cloud geometry and microphysics for the different simulations (see Table 1) were determined as follows. The first step in any ICA application is the assignment of a cloud model column to every pixel of a simulated radiance field respectively the geolocalization of the measured signal. The assignment is straightforward for nadir viewing directions but not at all trivial for slant viewing directions. For model data, the column corresponding to a certain pixel can be defined as the column where the first scatter event of a photon propagating backwards from the sensor into its viewing direction takes place. This approach is for example applied in the TICA DIR approximation by Gabriel and Evans (1996) and Wissmeier et al. (2013). Hence, 1000 photons were traced for every pixel and the column selected through the mean location of the first scatterings. In the case of measurements, geolocalization is possible from the known sensor position and viewing directions and information about cloud top height, which can be derived from the measurement data for example through stereographic methods. Volkmer et al. (2024) showed that the cloud top heights computed from the mean locations of the first scatter events correspond well to cloud top heights obtained with a stereographic retrieval. Thus, the chosen method for the column assignment through the locations of the first scattering is reasonable and consistent between model and measurement data.

Afterwards, the input parameters for every simulated pixel could be determined. The height of the mean locations of the first scatter events from the column assignment was used as cloud top height for every pixel. This height corresponds approximately to the height where the optical thickness reaches one as it was used for the cloud top height in model data in Scheck et al. (2018) and as it is obtained from measurements. As the clouds in the introduced approximations are defined such that their cloud top

**Figure 5.** Cloud geometry and microphysics for the parameterization. (a) Cloud top height. (b) Cloud surface orientation zenith angle. (c) Total optical thickness. (d) Ice fraction.

height is the height where the cloud water first exceeds zero, the height definition through an optical thickness of 1 or the location of the first scatter event is not completely consistent with that and the approximated clouds are shifted vertically relative to the real clouds. However, the deviation between the cloud top heights is small and affects only the amount of Rayleigh scattering above and within the cloud. Thus, it has only a negligible influence on the simulated radiances. In the future, this could nevertheless be accounted for, if necessary.




Once the cloud top height is known, the cloud surface orientation can be computed. It can be determined by fitting a plane through the three-dimensional points defining the cloud top height for neighboring columns as in (Scheck et al., 2018). Surface orientation angles can then be calculated from the normals of the fitted planes. Alternatively, a triangular mesh describing the three-dimensional surface of the cloud can be obtained through Poisson surface reconstruction as in Kölling (2020). The surface orientation zenith and azimuth angles are in this case derived from the triangular cloud surface mesh by computing the closest point on the surface mesh for the three-dimensional location of every pixel and using the respective surface normal. Here, the latter method was chosen as it is operationally applied to the specMACS measurements. Fig. 5(a) and (b) show the resulting cloud top height and surface orientation zenith angles. Cloud top heights are around 800 to 1000 m. The surface orientation zenith angle is close to zero in the center of the clouds and increases towards the edges.

Furthermore, the cloud fraction can be computed from a cloud mask as the ratio of the number of cloudy pixels to the total number of pixels. Cloudy pixels can either be defined as pixels where the optical thickness of the corresponding column in the model data exceeds a certain threshold value, or by a cloudmask from measurement data based on the brightness of the

pixels. Here, the brightness-based cloud mask for the specMACS instrument introduced in Pörtge et al. (2023) was applied to the simulated radiances. However, the results were very similar compared to the model based cloudmask. For the shown case, a single cloud fraction was computed from the cloud mask for the entire measurement range since the simulated cloud field is comparably homogeneous. For other cases with a more inhomogeneous distribution of clouds, only a subsection of the measurements surrounding the considered pixel should be used to compute the local cloud fraction for every simulated pixel. The cloud fraction defines the distance between the half-spherical cloud representing the cloud at the considered pixel and the surrounding half-spheres of the cloud field of the IDEFAX. Thus, the cloud fraction should represent this average distance.








Concerning the cloud microphysics for the radiative transfer simulations, the total vertical optical thickness of the respective column in the model data was used as optical thickness as it is common in ICA. For the determination of the ice fraction, the scatter type of the first scatter event was saved for every pixel besides the location of the first scatter events for the column assignment. The ice fraction was computed as the ratio of the number of scatterings on ice to the total number of scatterings. Similar results are obtained by simply taking the mean ice fraction of the grid boxes where the first scatter events take place for every pixel. Since the polarization signal is dominated by single scattering, this definition of the ice fraction through the first scattering was chosen. To exclude biases and uncertainties due to this definition, the analysis was repeated for pure liquid clouds as described below. Fig. 5(c) and (d) display the total optical thickness and the ice fraction for every pixel. The simulated clouds have total optical thicknesses of around 6 to 10. The ice fraction is smallest in the centers of the clouds due to the liquid at cloud top and increases for lower cloud top heights towards the edges of the clouds. All other parameters (such as e.g. the effective radii) were kept as described in Section 3 for the three-dimensional simulation.

Figure 6 shows radiative transfer simulation results for the realistic three-dimensional clouds (left column), the IDEFAX (middle column) and for the one-dimensional approximation (right column) for the total reflectivity (upper row) and polarized reflectivity (lower row). Visually, the results for the IDEFAX are much closer to the full three-dimensional simulation than the simulation using the one-dimensional ICA. The simulations based on the IDEFAX show lower values at the areas facing away from the sun as well as enhancements at the parts of the clouds that are oriented towards the sun, which is coming from the upper left. This cannot be covered by one-dimensional clouds. In addition, also the polarization signal is closer to the full 3D simulations for the IDEFAX with a more pronounced cloudbow and a less smooth signal than in the onedimensional approximation. The IDEFAX is also quantitatively closer to the three-dimensional simulation compared to the one-dimensional results. Fig. 7 shows scatter plots of the simulations using the IDEFAX and the plane-parallel ICA against the full 3D simulations for I and Q, as well as linear fits. The linear fits are closer to the identity for the IDEFAX and the correlation coefficients are higher than the correlation coefficients of the 1D simulation. The improvement of the agreement from the one-dimensional approximation to the IDEFAX is larger for I than for Q. This can be expected since the polarization signal is dominated by single scattering and therefore less affected by three dimensional radiative effects compared to the intensity. The difference of the correlation coefficients between the plane-parallel ICA and the IDEFAX is relatively small. The reason for this is, that clear-sky pixels without clouds or with very small optical thickness values were included in the analysis. These pixels are represented by the large number of pixels with small reflectivity in Figure 7. The difference between the plane-parallel ICA, the IDEFAX, and full 3D radiative transfer simulations for these pixels is small, leading to increased

**Figure 6.** Simulation results for realistic three-dimensional clouds in (a,d), the IDEFAX in (b,e) and the plane-parallel ICA in (c,f). (a,b,c) show reflectivity  $R_I$  and (d,e,f)  $R_Q$ . The dashed lines indicate the scattering angles.

correlation coefficients. If only cloudy pixels are considered, correlation coefficients of 0.52 and 0.70 are obtained for  $R_I$  and 0.89 and 0.90 for  $R_Q$  for the plane-parallel ICA and the IDEFAX, respectively. The increase of the correlation coefficients for  $R_I$  for cloudy pixels is significant. Overall, the addition of three parameters (cloud surface orientation zenith and azimuth angles, and cloud fraction) for simulations with half-spherical clouds, in addition to cloud top height and optical thickness as for 1D clouds, shows significant improvement towards full three-dimensional clouds and is simple.




The same analysis as discussed in the previous section was repeated for a cloud field consisting of pure liquid clouds to exclude potential uncertainties from the computation of the ice fraction for mixed phase clouds. For that, the clouds of the WRF cloud field were converted to entirely liquid clouds conserving the total optical thickness for every grid cell and the cloud geometry. The analysis for pure liquid clouds showed very similar results (not shown) compared to the mixed phase clouds which further validates the introduced parameterization of 3D cloud geometry.

In addition to the IDEFAX method itself, the neural network forward operators were validated against full 3D radiative transfer simulations with MYSTIC in a similar way. Figure 8 shows radiative transfer simulations for the green wavelength channel of specMACS for the WRF cloud field using the plane-parallel ICA or the IDEFAX similar to Figure 6 but obtained with the neural networks. The simulations with the neural networks look very similar to the Monte Carlo radiative transfer simulations. In addition, Figure 9 shows the corresponding scatter plots as in Figure 7 for the neural network forward operators. The results of the neural network forward operators are very similar to the results from the radiative transfer simulations above indicating that the training was accurate and the prediction errors of the neural networks are small. Small deviations between the

Figure 7. Scatter plots of the simulations with half-spherical clouds in the IDEFAX and one-dimensional clouds in the ICA against the full three-dimensional simulation for  $R_I$  (a,c) and  $R_Q$  (b.d). Panels (a) and (b) show the comparison for half-spherical clouds, (c) and (d) for one-dimensional clouds. The red curve shows the idenity, the orange curve a linear fit to the data. The fit parameters and the Pearson correlation coefficients (PCC, corresponding to the R value) are given in the legends.

Monte Carlo simulations and the neural networks are expected due to the random noise present in the Monte Carlo simulations.

#### 7 Discussion and conclusions



Three-dimensional radiative effects are important, but often neglected due to computational reasons. In this work, a fast forward operator for polarized 3D radiative transfer in the visible wavelength range was introduced. For this, a new approximation of three-dimensional cloud geometry for polarized radiative transfer in the visible wavelength range, IDEFAX, was developed. Different cloud geometry approximations were tested on their ability to reproduce 3D radiative effects. Three-dimensional radiative effects were best reproduced by an independent column approximation where each simulated pixel is represented by a field of half-spherical clouds defined by the surface orientation zenith and azimuth angles and the cloud fraction. The approximation of 3D cloud geometry with the IDEFAX has only three additional parameters defining the cloud geometry compared to the plane-parallel ICA. It is simple enough to allow for tabulation of simulated radiances such that 3D cloud geometry can

Figure 8. Simulation results of I and Q for the green color channel of specMACS with the neural network trained for 1D clouds in (c,f) and for half-spherical clouds with the IDEFAX in (b,e). The simulations correspond to the radiative transfer simulations in Figure 6 and use the same input parameters for every pixel. The dashed lines indicate the scattering angles.

directly be accounted for in retrievals. Besides the cloud fraction of the considered cloud field, only local information is needed for every column. The implementation of the IDEFAX is thus as simple as for the plane-parallel ICA.




A forward operator based on a simple look-up-table would have been computationally expensive due to the large number of parameters defining cloud and viewing geometry. Thus, the look-up-table was replaced by artitifical neural networks for both, the IDEFAX as well as the plane-parallel ICA. The neural network forward operators allow for very fast inference of the first three Stokes vector components I, Q, and U with only small memory requirements and can directly be used for retrievals. They are constructed to be used for (multi-angle polarimetric) retrievals with the polarization-resolving cameras of the specMACS instrument could however also be adapted for other polarimetric measurements. The prediction errors of the trained neural networks are small compared to typical measurement uncertainties of polarimetric instruments.

The IDEFAX and the neural network forward operators were validated against 3D radiative transfer simulations with MYS-TIC for a realistic field of low-level Arctic mixed-phase clouds simulated with the WRF model. There was a significant improvement towards full 3D simulations with the introduced IDEFAX compared to plane-parallel clouds in the classical ICA which are usually assumed for retrievals. Due to the implementation with neural networks, the computation time of radiances with the IDEFAX is comparable to the plane-parallel ICA and about five orders of magnitude faster than 3D radiative transfer simulations with MYSTIC for the shown example.

Figure 9. Scatter plots of the simulations with the neural networks for half-spherical clouds with the IDEFAX and one-dimensional clouds against the full three-dimensional simulation for  $R_I$  (a,c) and  $R_Q$  (b.d). Panels (a) and (b) show the comparison for half-spherical clouds, (c) and (d) for one-dimensional clouds. The red curve shows the idenity, the orange curve a linear fit to the data. The fit parameters and the Pearson correlation coefficients (corresponding to the R value) are given in the legends.

The introduced IDEFAX and forward operators are in principle also applicable to other multi-angle polarimetric measurements such as measurements by the Research Scanning Polarimeter (Cairns et al., 1999), AirHARP (Martins et al., 2018), SPEX airborne (Smit et al., 2019), AirMSPI (Diner et al., 2013), or POLDER (Deschamps et al., 1994). Potential further applications might include the field of data assimilation. For example, the fast forward operators used for the assimilation of satellite images (Scheck, 2021; Baur et al., 2023) so far only include the cloud top inclination correction by Scheck et al. (2018) for thick clouds and could be improved by applying the IDEFAX for optically thinner clouds. The number of input parameters of the neural networks could be extended and more dimensions included or parameter ranges adjusted to other polarimetric measurements, for new retrievals, or model applications.

A potential improvement of the neural network training (especially if more parameters are included) could be the application of Fourier feature mapping to improve the learning of high frequency features and reduce the spectral bias (Tancik et al., 2020). This could help to further reduce biases and errors especially for non-linear features such as the cloudbow and the non-linear dependence of the radiances on viewing and orientation geometry angles which are difficult to learn.

For the approximation itself and the validation some assumptions were made. The effective radii were set to constant values for the radiative transfer simulations with the WRF cloud field. Thus, the effect of spatial variations of effective radii was excluded here due to the focus on 3D cloud geometry. In the visible wavelength range, variations of the effective radius affect mostly the cloudbow which is highly sensitive to the cloud droplet size distribution. The cloudbow, however, is caused by single scattering and thus not strongly influenced by 3D radiative effects.






Moreover, the IDEFAX as well as the ICA with plane-parallel clouds assume homogeneously mixed clouds whereas the clouds obtained from the WRF simulations have a layered structure with liquid at cloud top and ice below as observed in typical Arctic mixed-phase clouds. To investigate the impact of this assumption, the validation in Section 6 was repeated for pure liquid clouds (not shown). The mixed-phase clouds were converted to liquid clouds by keeping the total optical thickness and thus the 3D cloud geometry conserved. The analysis with pure liquid clouds showed similar results. Thus, the additional uncertainty caused by the assumption of homogeneously mixed clouds and the definition of the ice fraction is small for the shown cloud case. The ice fraction here can be interpreted as an effective ice fraction of a homogeneously mixed cloud that has the same radiative effect as a layered cloud.

Finally, there could be an influence of the horizontal resolution with which the cloud surface orientation angles for the half-spherical clouds of the IDEFAX are computed. The triangular surface mesh used for the computation of the surface orientation angles in this study was constructed with a spatial resolution of 100 m. The simulated clouds were comparably smooth, but measurements of real clouds are generally more structured and the resolution used for the description of the cloud surface more important. To quantitatively investigate which spatial scales of cloud geometry variations dominate the 3D radiative effects, cloud simulations with higher horizontal resolution would be necessary. In addition, different instrument resolutions could also be studied in the future.

The introduced IDEFAX works well for the shown case of low-level Arctic mixed-phase clouds. In principle, the IDEFAX should be applicable to other cumuliform clouds whose 3D geometry resembles a half-spherical shape. On the other hand, the best agreement with the plane-parallel assumption was found for stratiform clouds (Di Girolamo et al., 2010). Thus, depending on the morphology of the observed cloud either a plane-parallel assumption in the classical ICA for very flat stratiform clouds or the IDEFAX for cumuliform clouds will be more appropriate. However, the quantification of the transition region between both cases remains challenging. Further validation studies are needed to test the IDEFAX for other cloud cases and investigate how generalizable it is.

Code and data availability. The 1D version of the radiative transfer solver MYSTIC is publicly available as part of libRadtran at http://www.libradtran.org
The 3D solver as well as the model weights for both neural networks can be provided upon request from the corresponding author. The
namelist for the WRF simulation is given in the supplement, the WRF simulation results are available from the authors upon request.

## Appendix A: Derivation of cloud geometrical thickness from the adiabatic cloud model

Following the adiabatic cloud model and Grosvenor et al. (2018), the liquid water content can be defined as:

$$LWC(z) = f_{ad}LWC_{ad}(z) = f_{ad}C_{w}(z - z_{base})$$
(A1)

with the adiabaticity  $f_{ad}$  and the condensation rate  $C_{w}$ , and z and  $z_{base}$  being the height and the cloud base height. Integrating this equation with height gives the liquid water path (LWP):

$$LWP = \frac{1}{2} f_{ad} C_w d^2 \tag{A2}$$

where  $d = z_{top} - z_{base}$  is the geometrical thickness of the cloud. On the other hand, the liquid water path is related to the optical thickness, assuming an extinction efficiency of 2 for geometrical optics, through:

$$\tau = \frac{3}{2} \frac{\text{LWP}}{\rho_{\text{w}} r_{\text{eff}}} \tag{A3}$$

Combining equations A2 and A3 and solving for d finally gives the formula for the geometrical thickness of a cloud as a function of the optical thickness using the adiabatic cloud model:

$$d = \sqrt{\frac{4}{3} \frac{\rho_{\rm w} r_{\rm eff}}{f_{\rm ad} C_{\rm w}}} \tau \tag{A4}$$

Author contributions. AW and BM developed the concept of this work. GK performed the WRF simulations and wrote the corresponding section. AW did the radiative transfer simulations, implemented the IDEFAX and the neural network forward operators, and wrote the manuscript with input from all co-authors. BM secured the funding.

Competing interests. Bernhard Mayer is member of the editorial board of AMT.

Acknowledgements. We would like to thank Philipp Gregor and Leonhard Scheck for their help concerning model setup and training as well as explanations to the fornado module. This research was supported by the German Research Foundation (DFG) within the project SPP 1294 under project number 442667104.

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
