# Peer review of "Parameterization of 3D cloud geometry and a neural-network-based fast forward operator for polarized radiative transfer"

_EGUsphere, 2025_

## Author Comment (AC1)

**Reply to referee #2**

We thank Referee #2 for reviewing the manuscript and the valuable comments and suggestions which we address below. The responses to the referee comments are given in blue italic letters.

The paper by Weber et al. addresses a very important research topic: the accurate and efficient simulation of atmospheric radiation fields for complex cloud scenarios (including polarisation).

Such radiative transfer models are important for the interpretation of remote sensing measurements and for the quantification of the atmospheric radiation budget. So far, usually 1D models are applied, which assume horizontally homogenous atmospheric properties. For many scenarios, they can not properly describe the atmospheric radiation fields.

The authors of this paper introduce a new method to describe the effects of 3D clouds using a simplified parameterisation for 3D clouds (half spherical clouds). Moreover, they also developed a fast neural network, which is trained on the simulations using the half spherical cloud parameterisation. Both new models are evaluated against a full 3D radiative transfer model, and a (slight) improvement with respect to the 1D cloud simulations was found.

The paper is scientifically sound and within the scope of AMT. However, the writing is sometimes confusing and has repetitions; also some quantities are not clearly defined (see also the minor points below).

After addressing my comments, the paper might be published in AMT.

**Major comments:**

1) While I see the benefits of the choice of the half sphere concept, it stays unclear why the authors have chosen this concept. The authors should motivate their choice and also discuss which other simple parameterisations might have been used instead. They might also discuss which more sophisticated concepts might be useful in improved future applications.

*Thank you very much for the comment. A discussion about other tested simple parameterizations and a motivation of our choice is given in Section 4, which we further extended:*

*"In a next step, different cloud geometries were tested as approximations to the real three-dimensional clouds. This included box clouds to account for the finite cloud size, tilted clouds to approximate the effect of cloud top inclination, and half-spherical clouds that include finite cloud size as well as cloud top inclination … A comparison of simulated Stokes vectors obtained with the different approximations against full 3D radiative transfer simulations of the realistic cloud field from the WRF model showed, that neither the finite cloud size (represented by the box clouds) nor the surface orientation (represented by the tilted clouds) alone are sufficient to capture the basic 3D radiative effects.*
*The cloud model that showed the best agreement with the full 3D cloud field was a field of half-spherical clouds as shown in Fig. 3(b). … An isolated half-spherical cloud not embedded into a cloud field showed less agreement with full 3D radiative transfer simulations than the field of half-spherical clouds defined through the cloud fraction."*

*We do not see a simple improvement of the method that could be integrated soon in the future. To further improve the results much more non-local information would have to be included. Additional parameters have to be chosen very carefully, make the computations more expensive, and the training of e.g. a neural network much more difficult. Especially the generation of training data would be very computationally expensive. The reason for the seemingly small improvement between plane-parallel ICA and IDEFAX is explained below.*

2) Several important aspects are not clear:

-are the clouds in a pixel represented by one half spherical cloud (as stated in line 6) or by a field of half spherical clouds (as stated in line 63)?

*The clouds in a pixel are represented by a field of half-spherical clouds. Line 6 and the following sentence together should make this clear:*

*"However, every column is approximated by an independent 3D half-spherical cloud instead of a plane-parallel homogeneous cloud. The half-spherical cloud is defined by the local cloud surface orientation angles and embedded in a cloud field with a given cloud fraction."*

-what is the size of one pixel? Does the size depend on the measurement properties and/or the grid of the RTM simulations?

*The resolution of the radiative transfer simulations was about 10m similar to the specMACS measurements, the WRF simulations have a resolution of 100m (see lines 84, 136). If another instrument with a different resolution would be simulated, the pixel size would change accordingly. We added additional sentences to Section 3 and 4 to make this clearer:*

*"The radiative transfer simulations have the same spatial resolution of about 10m as the measurements." "In this work, the pixel size was about 10m since this is the spatial resolution of the measurements of specMACS. For other applications, the pixel size could be chosen differently."*

3) The improvement of the new-methods over the 1D cloud approximation is rather small (Fig. 7): The correlation coefficient increases from 0.81 to 0.87 and from 0.85 to 0.86, respectively.

With such a small improvement, it remains unclear why a user should take the effort and use the new-methods instead of 1D cloud models.

The authors should explore, why the improvement is so small, and which model improvements (see my first comment) could be applied to increase the agreement to the full 3D simulations.

*It is true that the difference between the correlation coefficients is rather small. However, the slope and offset of the regression curves of IDEFAX are significantly closer to the identity line and visually the results in Figure 6 also look much better. The reason for the small difference between the correlation coefficients is, that all pixels including non-cloudy pixels were used to create the scatter plots, compute the correlation coefficients, and the regression lines. The cloud fraction of the simulated observations was however only about 0.4. Pixels without clouds or only very small optical thickness values are represented by the large number of pixels with very small reflectivity in the scatter plots. They lead to increased correlation coefficients in all cases. Without clouds there is of course only a negligible influence of 3D radiative effects due to 3D cloud geometry and the differences between all simulations are small. If only cloudy pixels are used to compute the correlation coefficients, much larger improvements are obtained.*

*The correlation coefficients for I for the plane-parallel ICA and the IDEFAX are then 0.52 and 0.70, respectively, which is an improvement of about 40%. For Q, the correlation coefficients are still comparably similar with 0.89 and 0.90 for cloudy pixels only. The smaller improvement for polarization compared to the total intensity is however expected, as polarization is much less affected by 3D radiative effects. We added an additional discussion about this to Section 6.*

*"The difference of the correlation coefficients between the plane-parallel ICA and the IDEFAX is relatively small. The reason for this is, that clear-sky pixels without clouds or with very small optical thickness values were included in the analysis. These pixels are represented by the large number of pixels with small reflectivity in Figure 7. The difference between the plane-parallel ICA, the IDEFAX, and full 3D radiative transfer simulations for these pixels is small, leading to increased correlation coefficients. If only cloudy pixels are considered, correlation coefficients of 0.52 and 0.70 are obtained for $R_I$ and 0.89 and 0.90 for $R_Q$ for the plane-parallel ICA and the IDEFAX, respectively. The increase of the correlation coefficients for $R_I$ for cloudy pixels is significant."*

**Minor points:**

-line 24/25: what is meant with roughening or smoothening of the brightness field? Could you add a suitable reference?

*We added a reference as suggested.*

-line 39: what is meant here with ‚high' and ‚low' resolution? Could you give numbers; maybe add a reference?

*We added information about the spatial resolution from the reference Cornet et al. 2010, which is referred to in the sentence.*

-line 55: what are the wavelength ranges for the three channels? What is the spatial resolution? Please add this information.

*We added more information about the instrument and a reference.*

-line 61: the ‚forward operator' should be introduced / defined; maybe a new subsection could be inserted starting with line 61?

*In line 61, a new paragraph is started. In addition, we added additional information about the forward operator.*

-line 103: what is meant with ‚the domain visible in the radiative transfer simulations?

(also for the figure caption of Fig. 6)

*The polarization-resolving cameras of specMACS have a slightly sideward looking viewing geometry along flight track. The black dashed lines in Figure 1 just indicate the part of the cloud field from the WRF simulations that is visible later in the radiative transfer simulations in Sections 3 and 6. The radiative transfer simulations were performed for an example observation of specMACS.*
*The cloud field obtained from WRF is larger than what is shown in Figure 1. Figure 1 shows only the relevant part for the radiative transfer simulations shown in Sections 3 and 6. We tried to reword the sentence.*

-Fig. 3: the pixel size should be graphically indicated in the figure. Is a cloud in a pixel represented by one half spherical cloud or by a field of half spherical clouds?

*Figure 3 represents only one pixel and the pixel size can vary depending on the application (see answer to comment 2 above). Every individual pixel is approximated by a field of half-spherical clouds. We added more details to Section 4 and tried to rewrite parts of it to make this more easily understandable.*

-Fig. 3 and text: what is the procedure if one cloud covers several pixels?

*The IDEFAX is based on the independent column approximation. Hence, every pixel is treated independently. So, it does not matter whether one cloud covers several measurement pixels or not. The pixel that should be simulated is in any case approximated by a field of half-spherical clouds defined through the properties of this pixel. This cloud field is of course internally made up of many pixels in the radiative transfer simulations. The internal grid size is independent of the simulated pixel size. We added an explanation to Section 4 to make this clearer.*

-Table 1 and text: please make clear for which area the cloud fraction defined? Is this done on a pixel basis, or for a larger area?

*The cloud fraction was computed from the entire image using a cloud mask since the cloud size and distribution of clouds within the cloud field is relatively homogeneous. If the cloud fraction/distribution of clouds would be very variable within an observation, only a part of the observations surrounding every considered pixel should be used to compute a cloud fraction. We added an additional discussion:*

*"For the shown case, a single cloud fraction was computed from the cloud mask for the entire measurement range since the simulated cloud field is comparably homogeneous. For other cases with a more inhomogeneous distribution of clouds, only a subsection of the measurements surrounding the considered pixel should be used to compute the local cloud fraction for every simulated pixel. The cloud fraction defines the distance between the half-spherical cloud representing the cloud at the considered pixel and the surrounding half-spheres of the cloud field of the IDEFAX. Thus, the cloud fraction should represent this average distance."*

---

## Author Comment (AC2)

**Reply to referee #1**

We thank Referee #1 for reviewing the manuscript and the valuable comments and suggestions which we address below. The responses to the referee comments are given in blue italic letters.

**General comments:**

This paper does more than what is described in its title. A whole new approximation to 3D radiative transfer (RT) in clouds at solar wavelengths is introduced, going by the whimsical name of IDEFAX (InDEpendent column local halF-sphere ApproXimation). It is an alternative to the well-known ICA (Independent Column Approximation) with 3 extra parameters to accomodate broken cloud fields. However, even these approximate 3D RT models are too cumbersome to use operationally due to their large parameter spaces, hence look-up tables (LUTs), with 11 and 14 dimensions, respectively, for ICA and IDEFAX. Therefore, the authors invoke a neural net (NN) model trained to accelerate either ICA or IDEFAX.

Both the new (IDEFAX) and old (ICA) approximations are tested against synthetic clouds and observations using the NCAR Weather Research and Forecasting (WRF) model followed by detailed computational 3D RT using the LMU MYSTIC code to estimate the intensity (Stokes I) and polarized (Stokes U and Q) radiances. The present application is for low-level mixed-phase Arctic clouds and the simulated observations are for LMU's airborne specMACS sensor.

The research is new and timely, and the paper is well-written. In this reviewer's opinion, it can be published in AMT after a revision that addresses the following questions.

**Specific comments:**

The most innovative part of this work is the IDEFAX model described in Section 4 and validated against high-fidelity (WRF+MYSTIC) vector 3D RT simulations in Section 6. That should be emphasized rather than the (more and more common) NN implementations in the revised title.

*We completely see your point and changed the title to focus more on the parameterization than the neural networks. In addition, we slightly adapted the introduction. The title reads now: Parameterization of 3D cloud geometry and a neural-network-based fast forward operator for polarized radiative transfer in low-level Arctic mixed-phase clouds*

Also, we hear about the ~5 orders-of-magnitude speedup of the NN models compared to MYSTIC. However, the more relevant speedup factors are IDEFAX or ICA vs MYSTIC and the NN implementations vs (LUT-based) IDEFAX and ICA. Please provide.

*Thank you very much for noting that. We added approximate times for the different options. The final computations were performed on our cluster which consists of a number of different machines with unfortunately quite different computation times for the same tasks. Thus, only approximate computation times are given.*

I may have missed this, but we'd like to know exactly how many WRF realizations of the Arctic clouds were used in the NN training. It feels like there is only one, which I doubt.

*We used only one realization of WRF for the validation of the IDEFAX and plane-parallel ICA against full 3D radiative transfer simulations. The neural networks were not trained with WRF simulations, but by randomly sampling all input parameters (summarized in Table 1) within their boundaries. Otherwise, the generation of training data would have been very computationally expensive.*

Lastly, does the new cloud fraction parameter not have an upper limit that is less than unity? 78.5% for hemispheres on a square cartesian grid, or 81.4% for closely packed hemispheres.

*Yes, this is true, the cloud fraction has an upper limit less than unity. We added an additional sentence about this to Section 4.*

**Technical corrections:**

p. 1, l. 9: Remove 2nd coma.

*Changed as suggested.*

p. 1, l. 25: either roughening and smoothing --> both roughening and smoothing
    or --> either roughening or smoothing

*Changed as suggested.*

l. 30: most retrievals --> most operational retrievals

*Changed as suggested.*

l. 75: Word "used" is unnecessary.

*Changed as suggested.*

l. 111, and many times after this: 1000m --> 1000 m, with unbreakable space before unit

*We added a space between numbers and units throughout the paper.*

Fig. 2: Would be beneficial to show lines of equal scattering angle here and in similar figures.

*We added scattering angles to Figures 2, 6, and 8 as suggested.*

l. 146: Need either comas or parentheses or both in "radiances respectively Stokes vectors".

*Changed as suggested.*

l. 271: Remove first "the".
*Changed as suggested.*

l. 300, 1st sentence: Best to separate clauses with a coma. Elsewhere too, e.g., often before "which" (unless it should be "that").

*Changed as suggested. Still missing comas will hopefully be corrected during typesetting.*

l. 325, and elsewhere: Referring to "real" clouds is misleading, better to use "realistic" or "WRF".

*Changed as suggested.*

---

## Author Response (AR2)

**Reply to referee #1**

We thank Referee #1 for reviewing the manuscript and the valuable comments and suggestions which we address below. The responses to the referee comments are given in blue italic letters.

The authors have made all the edits that I suggested. However, when copy/pasting my draft comments into the AMT dialog box, these last 4 sequential corrections were left behind. Apologies for that.

Fig. 7, caption: (corresponding to the R value) --> (PCC, corresponding to the R value)

changed as suggested

Fig. 8, caption: Probably should be (b,d) --> (c,f) and (a,c) --> (b,e). No?

changed as suggested

I. 389 (original submission): reduces --> reduce

changed as suggested

I. 427 (original submission): z --> z \text{top} in view of the definition of z in Eq. (A1)

changed as suggested
* * *
Also, with respect to my comment:

"I may have missed this, but we'd like to know exactly how many WRF realizations of the Arctic clouds were used in the NN training. It feels like there is only one, which I doubt."

I suggest that the authors better clarify how the NN sampling is done over the model parameters using a single run of WRF followed by MYSTIC.

We added more information about the generation of training data to the corresponding section.
* * *
Finally, putting "the" in front of every mention of IDEFAX sounds strange to me, although I can see the logic. If it were ICA, it sounds right because we tend to read it as "the I-C-A", not the word "ica". I think it is because the clever new acronym is long enough and pronounceable, just like a real word, So, it doesn't require the "the" every time. Maybe if you write it as "the IDEFAX model", it would be OK. Anyway, the English language experts will decide.

A native American at our institute told us that he would put a "the" in front of IDEFAX. But as non-native speakers we are definitely open for correction and would let the English language experts decide as suggested.

**Reply to referee #2**

We thank Referee #2 for reviewing the manuscript and the valuable comments and suggestions which we address below. The responses to the referee comments are given in blue italic letters.

Almost all of my comments were well addressed, but I am still not happy with the reply to my previous comment:

- 2) Several important aspects are not clear:
- -are the clouds in a pixel represented by one half spherical cloud (as stated in line 6) or by a field of half spherical clouds (as stated in line 63)?

It seems the authors did not change the respective parts of the text (line 63 is now line 66), and I still find this inconsistent and confusing

We are very sorry that we did not address your previous comment sufficiently. We did now change the wording in the abstract to make it clearer that a field of half-spherical clouds is used. In addition, we slightly restructured parts of the introduction. We hope the new description is now more understandable and less confusing.

The corresponding section of the abstract reads now: "The basic idea behind this approximation is similar to the independent column approximation assuming plane-parallel clouds. However, every column is approximated by an independent field of 3D half-spherical clouds instead of a plane-parallel homogeneous cloud. This field of half-spherical clouds is defined by the local cloud surface orientation angles and the cloud fraction."